# Human Mitochondrial Pathologies of the Respiratory Chain and ATP Synthase: Contributions from Studies of *Saccharomyces cerevisiae*

**DOI:** 10.3390/life10110304

**Published:** 2020-11-23

**Authors:** Leticia V. R. Franco, Luca Bremner, Mario H. Barros

**Affiliations:** 1Department of Biological Sciences, Columbia University, New York, NY 10027, USA; lib2109@columbia.edu; 2Department of Microbiology, Institute of Biomedical Sciences, Universidade de Sao Paulo, Sao Paulo 05508-900, Brazil; mariohb@usp.br

**Keywords:** mitochondrial diseases, respiratory chain, yeast, *Saccharomyces cerevisiae*, *pet* mutants

## Abstract

The ease with which the unicellular yeast *Saccharomyces cerevisiae* can be manipulated genetically and biochemically has established this organism as a good model for the study of human mitochondrial diseases. The combined use of biochemical and molecular genetic tools has been instrumental in elucidating the functions of numerous yeast nuclear gene products with human homologs that affect a large number of metabolic and biological processes, including those housed in mitochondria. These include structural and catalytic subunits of enzymes and protein factors that impinge on the biogenesis of the respiratory chain. This article will review what is currently known about the genetics and clinical phenotypes of mitochondrial diseases of the respiratory chain and ATP synthase, with special emphasis on the contribution of information gained from *pet* mutants with mutations in nuclear genes that impair mitochondrial respiration. Our intent is to provide the yeast mitochondrial specialist with basic knowledge of human mitochondrial pathologies and the human specialist with information on how genes that directly and indirectly affect respiration were identified and characterized in yeast.

## 1. Introduction

Mitochondria are dynamic organelles that supply most of the ATP needed to sustain the different energy-demanding activities of eukaryotic cells. Their ATP generating pathway consists of the oxphos complexes—four hetero-oligomeric complexes that make up the electron transfer chain plus the ATP synthase. Some complexes of this pathway are genetic hybrids composed of both mitochondrial and nuclear gene products. Most of the organelle, however, consists of proteins that are encoded by nuclear genes, their mRNAs translated on cytoplasmic ribosomes and the protein products transported to their proper internal membrane and soluble compartments. It is estimated that the nuclear proteome of *Saccharomyces cerevisiae* dedicated to the maintenance of respiratory competent mitochondria consists of at least 900 proteins [1]. Much of our information about the functions of this class of nuclear genes has been learned from studies of yeast. The identity of proteins localized in mitochondria and the phenotypic consequences of null mutations in their genes has come from large-scale proteomic studies. Information about the functions of nuclear gene products at the molecular level, however, has been gathered from a large body of previously known information of protein structure and function and from more recent in-depth biochemical analyses of yeast nuclear *pet* mutants. A search of the current Saccharomyces Genome Database (SGD) [2] indicates that the functions of some 170 proteins of the mitochondrial proteome essential for respiration and/or ATP synthesis are still not known.

## 2. Yeast, a Model for Studying Mitochondrial Function and Biogenesis

The last 50 years have witnessed unparalleled technical advances in deciphering the genetic compositions of whole genomes, so much so that whole new specialties have been born with the goal of developing tools for analyzing and dealing with this wealth of data in almost every major area of biological research. Of course, genes are only a starting point for the more interesting question of what their protein products do. This is one of the central questions of proteomics, which strives to develop methods for the simultaneous analysis of the entire complement of proteins in organisms, tissues and cells. Although the field as it stands today is highly successful in many important areas, such as ascertaining subcellular protein localization, their transient, as well as stable, physical interactions and patterns of expression during cell division, development, and diseased states, an understanding of their molecular functions and the specific cellular process they participate in continues to depend on slugfest genetics and biochemistry on a single or a small number of genes.

Respiratory deficient *pet* mutants of *S. cerevisiae*, particularly those obtained in Alexander Tzagoloff’s laboratory [3], have been helpful in identifying a number of nuclear gene products essential for maintaining structurally and functionally competent mitochondria. The genes represented by about two thirds of the 215 complementation groups in such collections [3] have been characterized and their functions deduced.

One of the unexpected finding to have emerged from the functional analyses of *pet* mutants is the large extent to which expression of mitochondrial genes depends on mRNA-specific factors encoded in nuclear DNA. Also unexpected are the many accessory proteins that function in translation and assembly of the respiratory and ATP synthase complexes. For the most part this class of mitochondrial proteins target translation of specific mitochondrial mRNAs and maturation and biogenesis of their encoded proteins. For example, some three dozen proteins that are not constituents of cytochrome oxidase are currently known to be required specifically for the assembly of this single respiratory complex. Foreseeably, still unrecognized assembly factors may be discovered with further biochemical and genetic screens of uncharacterized *pet* mutants.

Mutations in human mitochondrial genes for subunit polypeptides of NADH-coenzyme Q reductase, cytochrome oxidase (COX, Complex IV), coenzyme Q-cytochrome *c* reductase (bc1 complex, Complex III) and ATP synthase (Complex V), were the first to be identified [4]. Subsequently, mutations presenting different clinical phenotypes were reported in nuclear genes that code for protein subunits of the ATP synthase and factors that function as chaperones during its assembly [5,6,7] and enzymes of biosynthetic pathways for heme *a* [8] and coenzyme Q [9]. A more complete blueprint of the regulatory proteins and chaperone factors that contribute to the biogenesis of respiratory competent mitochondria will uncover new chaperones and regulatory factors, some of which will undoubtedly have human homologs.

Human cells are more complex than yeast cells; and the same can be stated about the mitochondria of these two organisms. The higher the organization and the complexity, the more the consequences of one given deficiency differ. In many tested mutations, the phenotypes observed in humans are more deleterious for cell survival than in the yeast counterpart, which is not only true because yeast can ferment but also because of the variable energy demand of a complex organism with different tissues. As an example, the deletion of *MRX10* in yeast did not impair its respiratory capacity but mutations in the human counterpart led to respiratory impairment even in cells with low energetic demand such as fibroblasts [10]. In other circumstances, due to the need of proper protein-protein interactions, or just because of evolutionary divergence, the possibility of heterologous complementation is lost. For instance, yeast *shy1* mutants are not complemented by the human homolog *SURF1*, even with chimeric versions of the gene [11]. However, when the human genes do not complement the respective yeast mutant, it is still possible to evaluate the pathogenicity of a given mutation by constructing an allele with the corresponding change in the yeast gene.

## 3. Strategy for Determining the Function of Unknown Mitochondrial Proteins

A useful initial step for identifying the biochemical lesion of *pet* mutants is to assign them to one of three broad phenotypic classes based on the spectral properties of their mitochondria. This substantially reduces the number of subsequent assays. For example, a strain showing a normal complement of mitochondrial cytochromes and respiratory chain complexes can be excluded from harboring a mutation in a gene that affects mitochondrial translation, as both COX and the bc1 complex contain subunit polypeptides (cytochromes *a*, *a*_3_, and cytochrome *b*) that are translated on mitochondrial ribosomes [12]. By the same token, a selective loss of cytochrome *a* generally signals a mutation in a gene required either for:
(1)Expression of one of the mitochondrial gene products of the COX complex;(2)Assembly of this complex; or(3)Maturation of the heme active centers of the enzyme.


A similar argument can be made for mutants lacking cytochrome *b*, except that a safe assumption is that the lesion affects some aspect of bc1 biogenesis.

Finally, mutations in genes that are directly or indirectly required for the maintenance of mitochondrial DNA (mtDNA), undergo a large deletion or complete loss of the genome resulting in a population of cytoplasmic petites (ρ- and ρ^0^ mutants). Particularly prevalent in this class are mitochondrial protein synthesis mutants, (e.g. aminoacyl tRNA synthases and ribosomal mutants) [13] and mutants with defective ATP synthase [14,15]. Both mitochondrial translation and ATP synthase mutants display the absence of “*a*” and “*b*” type cytochromes for the reasons indicated above. An outline of the screens useful in identifying different classes of *pet* mutants is shown in Figure 1.

## 4. Pathological Mutations in Respiratory Chain and ATP Synthase Human Genes with *S. cerevisiae* Homologs

In this section, the reader will find information on complexes II, III and IV, ATP synthase, Coenzyme Q and cytochrome *c*. All genes with pathogenic variants encoding subunits of the respiratory complexes and ATP synthase will be described. Regarding assembly factors, we have included only the ones that have yeast homologs.

### 4.1. Complex II

Succinate-coenzyme Q oxidoreductase, or Complex II, is a membrane-bound enzyme that functions in both the TCA cycle and the electron transfer chain. In the TCA cycle, it catalyzes the oxidation of succinate to fumarate using coenzyme Q as the electron acceptor without accompanying ATP synthesis [16]. The reduced coenzyme Q formed in this reaction is then reoxidized by Complex III, a reaction coupled to the translocation of protons across the mitochondrial inner membrane and ATP synthesis. Fungal and mammalian Complex II is embedded in the mitochondrial inner membrane, with a large portion protruding into the matrix. It is composed of four protein subunits, including the flavoprotein succinate dehydrogenase with covalently bound FAD and the iron sulfur subunit. Both of these catalytic subunits are peripheral proteins facing the matrix side of the inner membrane [17,18]. All four subunits of Complex II are encoded in the nucleus. The two catalytic subunits of Complex II are encoded by *SDHA* and *SDHB* in humans and by *SDH1* and *SDH2* in yeast. The other two subunits are integral membrane proteins that form a dimer that houses a single heme *b* group of cytochrome *b*_560_ and the two coenzyme Q binding sites of the complex. These two membrane anchors of the catalytic sector are encoded by human *SDHC* and *SDGD* and yeast *SDH3* and *SDH4*.

In yeast, Sdh3 is a bifunctional protein that is also a subunit of the TIM22 protein translocase complex responsible for transporting and integrating members of the substrate exchange carrier family into the inner membrane [19]. Electrons released during the oxidation of succinate first reduce the FAD cofactor of SDHA and are then sequentially transferred to three iron-sulfur clusters in SDHB before reacting with coenzyme Q [18,20,21,22]. *S. cerevisiae sdh1–4* mutants are respiratory deficient and display a severe growth defect on non-fermentable carbon sources such as glycerol and ethanol. The function of the cytochrome *b*_560_ is not fully understood, but it is thought to shuttle electrons between the two ubiquinone binding sites [23].

#### 4.1.1. Mutations in Complex II Catalytic and Structural Subunits

Patients with lactic acidosis resulting from reduced succinate dehydrogenase activity have been linked to mutations in all four gene products of human Complex II. Although patients with deficiencies in the respiratory chain complexes, including Complex II, had been reported earlier [24], *SDHA* was the first instance of a nuclear encoded protein of the electron transfer chain with a mutation shown to cause a respiratory defect [25]. In that study, two siblings were homozygous for an R554T substitution in *SDHA* which resulted in Leigh syndrome, a severe neurological disorder that affects the central nervous system first described by Denis Leigh in 1951 [26]. The attribution of the phenotype to the mutation in *SDHA* was confirmed when the homologous mutation in the yeast flavoprotein was shown to have a deleterious effect on Complex II activity [25]. In the past 20 years, other *SDHA* mutations have been reported in patients presenting different clinical phenotypes (Table 1).

Interestingly, the same homozygous G555E substitution was identified in patients with distinct phenotypes: Leigh syndrome [27] and neonatal isolated cardiomyopathy [28]. This mutation was also found in a baby that died at five months of age following a respiratory infection before developing other phenotypes [29]. There is evidence that the G555E mutation prevents an adequate interaction between SDHA and SDHB [29]. This is supported by earlier studies on yeast Complex II assembly involving chimeric human/yeast genes [30].

More recently, germline mutations in *SDHA* were found in three patients with persistent polyclonal B cell lymphocytosis. In contrast to the other cases mentioned, the mutations resulted in a substantial increase of Complex II activity, leading to fumarate accumulation which engaged the KEAP1–Nrf2 system to drive the expression of inflammatory cytokines [31]. Mitochondrial pathologies have also been ascribed to mutations in *SDHB* and *SDHD* (Table 1).

#### 4.1.2. Mutations in Complex II Assembly Factors

Respiratory deficiency is also elicited by mutations in accessory proteins that are required for assembly but are not constituents of Complex II [32,33]. Four such assembly factors have been identified for the human complex: *SDHAF1-4*, with yeast homologs *SDH6*, *SDH5*, *SDH7*, and *SDH8*, respectively [32,34,35,36]. Mutations in *SDHAF1* (yeast *SDH6*), that codes for an assembly factor of Complex II, result in infantile leukoencephalopathy and have been reported in five patients, some sharing substitutions at the same residues [34,37].

#### 4.1.3. Complex II and Paragangliomas

Mutations in *SDHA*, *SDHB*, *SDHC*, *SDHD*, *SDHAF2*, and *SDHAF3* have been identified in an increasing number of neoplasms, mainly paragangliomas (rare neoplasms of the autonomic nervous system) and gastrointestinal stromal tumors. A discussion of these mutations is beyond the scope of this article but the topic has been extensively reviewed by others [38,39,40,41,42,43]. Variants in these genes have been associated with a high probability of developing cancer. To improve the identification of putative cancer-inducing human genetic variants, 22 known human variants of *SDHA* were examined in yeast [44]. Complementation tests of the homologous mutations in yeast *sdh1* identified 16 variants that affected the growth of yeast on the non-fermentable carbon source glycerol. The corresponding 16 human variants were proposed to be putative cancer inducing amino acid substitutions.

### 4.2. Complex III

Complex III or the *bc1* complex is an integral inner membrane homodimeric complex of the mitochondrial inner membrane that catalyzes the oxidation of reduced coenzyme Q and reduction of cytochrome *c*, a reaction coupled to the translocation of protons from the matrix to the inter-membrane space [53]. Both human and yeast Complex III contain three catalytic subunits: a mitochondrially-encoded cytochrome *b*, with two non-covalently bound heme *b* containing redox centers corresponding to cytochromes *b*_H_ and *b*_L_; cytochrome *c*_1_, with a covalently linked heme *b*; and the Rieske iron-sulfur protein [53,54,55,56]. In addition to the three catalytic subunits, Complex III contains seven other subunits, four of which are essential for the assembly and stability of the complex but do not participate in either electron transfer or proton translocation (Table 2). Like cytochrome *c*_1_ and the Rieske iron sulfur protein, all the non-catalytic subunits are products of nuclear genes.

Assembly of Complex III also depends on nuclear encoded chaperones and on factors that regulate translation and assembly of cytochrome *b*. Most of the currently known Complex III assembly factors that were first described in yeast are conserved in humans (Table 2). Among yeast factors that have human homologs associated with diseases, there are Cbp6 and Cbp4. Cbp6, together with Cbp3, forms a complex with nascent apocytochrome *b* [57] for subsequent addition of heme to form the redox center at the cytochrome *b*_L_ site, followed by stabilization of the partially mature protein by Cbp4 [58] and further hemylation of the cytochrome *b*_H_ site [59]. Mitochondrial pathologies have also been reported in patients with mutations in human homologs of two other factors, Bcs1 and Mzm1, both needed for maturation and insertion of the Rieske iron sulfur protein into the complex [60,61,62].

Most laboratory strains of *S. cerevisiae* have a mitochondrial cytochrome *b* gene (*COB*) containing group I and II introns that are post-transcriptionally removed [63,64]. Some of the group II introns contain reading frames that code for factors, termed maturases, that function in splicing their own introns [65]. Splicing of the terminal group I intron is aided by a protein factor encoded by a nuclear gene [66]. In addition to these splicing factors, expression of *COB* depends on other factors that stabilize and activate translation of the mRNA [67,68]. Due to the absence of introns in the human cytochrome *b* gene and of 5′- non-coding sequences in the human mRNA, none of the yeast RNA splicing factors and translational activators have human homologs.

Complex III disorders are relatively rare but, like mutations in the other respiratory complexes, they present a wide spectrum of phenotypes. Complex III deficiency can be caused by mutations in the mitochondrially-encoded cytochrome *b*, in nuclear genes coding for catalytic and structural subunits and in ancillary proteins that function in assembly of the complex.

#### 4.2.1. Mutations in Complex III Catalytic Subunits

Most of the known Complex III associated pathologies result from mutations in *MT-CYB*, the human mitochondrial cytochrome *b* gene, that have so far been described in 49 different positions of the genome [69]. Typical phenotypes include MELAS (mitochondrial myopathy, encephalopathy, lactic acidosis, and stroke-like episodes), LHON (Leber’s Hereditary Optic Neuropathy), hearing loss and in some cases less severe phenotypes expressed in exercise intolerance (reviewed in [70]). While some cytochrome *b* mutations are maternally inherited, others are heteroplasmic and are present mainly in muscle tissue, suggesting that they arise de novo after differentiation of the primary germ layers. Additionally, some LHON patients with a mutation in one of the mitochondrially encoded Complex I genes have a second mutation in cytochrome *b*, which exacerbates the severity of the pathology [71,72,73]. Mutations in the nuclear *UQCR4* (yeast *CYC1*) and *UQCRFS1* (yeast *RIP1*) genes are much rarer and were more recently identified (Table 3).

#### 4.2.2. Mutations in Complex III Structural Subunits

A 4 bp deletion in *UQCRB* resulting in a change in the last seven amino acids and an addition of a stretch of 14 amino acids at the C-terminal end of the protein was the first described case of a Complex III deficiency resulting from mutations in a nuclear encoded subunit of the complex [74]. Interestingly, earlier studies on yeast Complex III showed that deletions in the C-terminal helical domain of the *QCR7* homolog resulted in reduced levels of the subunit and of cytochrome *b*, Rip1, and Qcr8 [75]. Taken together, these studies show the importance of the helical domain of *UQCRB* in the maintenance or assembly of human and yeast Complex III.

The discovery of the deletions in *UQCRB* was followed by the identification of a pathogenic variant of *UQCRQ* [76]. A pathogenic mutation in *UQCRC2* was also identified: three patients from a Mexican consanguineous family with neonatal onset of hypoglycemia and lactic acidosis were found to have a homozygous mutation leading to a R183W substitution in the Core 2 subunit. The patients also showed hyperammonemia, high urine organic acids and elevated plasma hydroxyl fatty acids, suggesting that the *UQCRC2* mutation may elicit secondary effects on TCA and urea cycles and beta-oxidation. Modeling of the crystal structure of bovine Complex III predicted that the R183W mutation would disrupt the hydrophobic interface of the UQCRC2 homodimer, leading to Complex III destabilization. This was supported by an 80% decrease of the complex in the patients, even though Complex III activity was only marginally affected [77]. More recently, the same homozygous mutation with similar symptoms was described in a French child [78]. Furthermore, the expression of *UQCRC2* is upregulated in multiple human tumors, whereas its suppression inhibits cancer cells and induces senescence [79].

#### 4.2.3. Mutations in Complex III Assembly Factors

Together with cytochrome *b*, most Complex III pathologies are derived from mutations in the nuclear *BCS1L* gene. Its yeast homolog *BCS1* is an AAA protease that was shown to be required for the expression/maturation of Rip1 [60,61]. Later studies indicated that Bcs1 promotes one of the steps in the translocation of Rip1 necessary for incorporation of the iron-sulfur center [80]. Bcs1 exists as a heptamer with a contractile central cavity that participates in the translocation of the folded Rip1. Additionally, Bcs1 is associated with the complex III assembly module and its dissociation ends the maturation process [81].

Mutations in *BCS1L* comprise a wide spectrum of pathologies, including: GRACILE syndrome (growth retardation, aminoaciduria, cholestasis, iron overload, lactic acidosis, early death) [82,83,84,85,86,87,88]; Björstand syndrome, characterized by hearing loss and *pili torti* [89,90,91,92,93,94]; encephalopathy [95,96,97,98]; lactic acidosis, liver dysfunction and tubulopathy [99,100,101,102]; muscle weakness, focal motor seizures and optic atrophy [103], among others. Besides low steady state levels of Complex III, cells from patients with *BSC1L* mutations have impaired mitochondrial import of the protein as evidenced by its accumulation in the cytosol [91]. An adult harboring a R69C missense mutation was diagnosed with aminoaciduria, seizures, bilateral sensorineural deafness, and learning difficulties. Yeast complementation studies corroborate that the R69C mutation impairs the respiratory capacity of the cell [104].

Assembly of Rip1 is thought to be the last step in the biogenesis of Complex III [81]. In addition to Bcs1, this assembly step was found to require the product of the yeast nuclear *MZM1* gene [105]. The first case of Complex III deficiency caused by a mutation in *LYRM7*, the human homolog of *MZM1*, was reported in 2013 [106]. The equivalent mutation in yeast resulted in decreased oxygen consumption as a result of reduced steady state levels of Rieske protein and Complex III. Since then, Complex III deficiency caused by *LYRM7* has been identified in patients presenting with leukoencephalopathy [107,108,109] and liver dysfunction [110].

Mutations have also been reported in recent years in the human *UQCC2* and *UQCC3* genes that code for protein homologs of yeast complex III assembly factors Cbp6 and Cbp4. Complex III deficiency was found in a patient with a homozygous mutation in *UQCC2* [111]. This study demonstrated that the biochemical phenotype produced by the *UQCC2* mutation is similar to that reported in yeast [57], as cytochrome *b* synthesis and stability was decreased in the patient’s fibroblasts [111]. A homozygous mutation in *UQCC2* leading to a Complex III deficiency was also reported in a consanguineous baby presenting neonatal encephalomyopathy. This mutation resulted in a secondary deficiency of Complex I [112]. The authors proposed that assembled Complex III is required for the stability or assembly of complexes I and IV, which may be related to supercomplex formation. Interestingly, a recent study [113] showed that the ND1 subunit of Complex I co-immunoprecipitated with newly synthesized UQCRFS1 of Complex III in mammalian mitochondria, indicating a possible coordination of the assembly of the two complexes.

During assembly of yeast Complex III, Cbp4 is recruited by the Cbp3–Cbp6-cytochrome *b* ternary complex following release of the latter from the mitoribosome [57]. Wanschers et al. [114] described a homozygous mutation in *UQCC3*, the human homolog of *CBP4*, in a patient diagnosed with isolated Complex III deficiency. Cultured fibroblasts from the patient were partially deficient in cytochrome *b* and had no detectable UQCC3 protein. The authors concluded that UQCC3 functions in Complex III assembly downstream of UQCC1 and UQCC2, as the absence of UQCC3 did not affect the levels of UQCC1 and UQCC2 [114]. These observations are consistent with the above mentioned sequential interaction of Cbp4 with the Cbp3-Cbp6-cytochrome *b* complex during synthesis and assembly of cytochrome *b* [57].

### 4.3. Complex IV

Complex IV or cytochrome oxidase (COX) is an integral mitochondrial inner membrane protein complex that catalyzes the oxidation of cytochrome *c* and the reduction of molecular oxygen to water. This reaction is coupled to the translocation of protons from the matrix to the inter-membrane space [117]. In both human and yeast mitochondria, the COX catalytic core is composed of three proteins encoded in the mitochondrial DNA. They are Cox1, Cox2, and Cox3 in yeast, and MTCO1, MTCO2, and MTCO3 in humans. All the redox centers of COX are located in the Cox1 and Cox2 subunits. Yeast but not human Cox2 is synthesized with a cleavable N-terminal presequence that is required for correct insertion of the protein into the membrane [118]. Cox3 does not contain redox centers. It is thought to stabilize the catalytic core and to enhance the uptake of protons from the mitochondrial matrix [119]. One of the redox centers of Cox1, corresponding to cytochrome *a*, contains heme *a*. The second center, corresponding to cytochrome *a*_3_, consists of a binuclear heme *a*-Cu_B_. The third center, located on Cox2, is the binuclear Cu_A_. In addition to the catalytic core, COX is composed of several other structural subunits, all encoded in nuclear DNA (Table 4). Recently NDUFA4/COXFA4, a subunit previously thought to be part of Complex I, has been shown to be a subunit of COX [120].

COX catalyzes the consecutive transfer of 4 electrons from cytochrome *c* to a molecule of oxygen bound to the heme *a* of cytochrome *a*_3_ with the formation of water. Each electron of cytochrome *c* first reduces the Cu_A_ center of Cox2, from which it is then transferred to the heme *a* of cytochrome *a* and finally to the binuclear heme *a* –Cu center of cytochrome *a*_3_ [121,122].

The heme *a* differs from heme *b* at two positions of the porphyrin ring. While heme *b* is present in hemoglobin and most heme containing enzymes, heme *a* appears only in COX. The biosynthesis of heme *a* is initiated by the addition of a farnesyl group to the C-2 position of the heme *b* porphyrin ring by farnesyl transferase, encoded by *COX10* [123]. The resulting heme *o* is then converted to heme *a* by the oxidation of a methyl to a formyl group on C-8 of the porphyrin ring in a reaction that requires Cox15, mitochondrial ferredoxin, and ferredoxin reductase [124,125]. Some other factors, such as Shy1 (human SURF1) and Pet117, have been shown to be involved in the hemylation of Cox1 [126,127]. Additionally, several proteins with human homologs, including Cox17, Sco1, Sco2, Cox11, Cox19, Cox23, Cox16, and Cmc1 have been implicated in the trafficking of copper and maturation of the Cu_A_ and heme *a*-Cu_B_ centers, as reviewed elsewhere [128].

Pathological mutations have been reported in both mitochondrial and nuclear encoded COX subunits, as well as in proteins involved in the biogenesis of the complex. As of today, mutations in more than 20 genes can lead to COX deficiency with a broad spectrum of clinical phenotypes (Table 5).

Furthermore, COX interacts with other complexes of the electron transport chain in entities called supercomplexes or respirasomes that are thought to provide a kinetic advantage by allowing for a more efficient transfer of electrons between the respiratory complexes and their intermediary carriers cytochrome *c* and coenzyme Q [129,130]. In humans, the respirasome is composed of Complex I, III, and IV in variable stoichiometry, while in yeast it is composed of Complex III and IV in strict 2:2 and 2:1 ratios [61,131,132]. Mutations resulting in COX deficiency affect respirasome biogenesis, which could ultimately lead to complex secondary phenotypes, as discussed elsewhere [133].

#### 4.3.1. Mutations in Complex IV Catalytic Subunits

Generally, when compared to nuclear structural subunits and factors, mitochondrial COX genes are associated with milder and late onset clinical phenotypes [135]. As of today, there are 42 pathogenic mutations reported for MTCO1, 26 for MTCO2, and 24 for MTCO3 [69]. The phenotypes associated with mutations in these subunits are briefly summarized in the paragraphs bellow and are cited from the MITOMAP [69].

The most frequent homoplasmic pathogenic mutations in *MTCO1* are associated with prostate cancer, LHON, SNHL (sensorineural hearing loss) and DEAF (maternally-inherited deafness). The most frequent diseases caused by homoplasmic variants are dilated cardiomyopathy and maternally inherited epilepsy and ataxia. Clinical phenotypes associated with heteroplasmic variants include epilepsy partialis continua, Leigh syndrome, asthenozoospermic infertility, MELAS, myoglobinuria, motor neuron disease, Rhabdomyolysis and acquired idiopathic sideroblastic anemia. Additionally, both homoplasmic and heteroplasmic variants can lead to exercise intolerance.

Homoplasmic pathogenic *MTCO2* variants are mostly associated with progressive encephalomyopathy, possible susceptibility to hypertrophic cardiomyopathy (HCM), SNHL, DEAF, and LHON. For example, there are 147 sequences containing the m.7859G>A substitution that causes progressive encephalomyopathy. Less frequent mutations can cause Alpers-Huttenlocher-like, Asthenozoospermia, developmental delay, ataxia, seizure, hypotonia, hepatic failure, myopathy, MELAS, cerebellar and pyramidal syndrome with cognitive impairment, pseudoexfoliation glaucoma, multisystem disorder, Rhabdomyolysis, biliary atresia, and MIDD (maternally-inherited diabetes and deafness).

The most frequent pathogenic *MTCO3* variants are homoplasmic and lead to LHON. Other clinical phenotypes associated with mutations in *MTCO3* include Alzheimer’s disease, MELAS, Leigh syndrome, cardiomyopathy, exercise intolerance, myoglobinuria, myopathy, asthenozoospermia, failure to thrive, cognitive impairment, optic atrophy, encephalopathy, rhabdomyolysis, and sporadic bilateral optic neuropathy.

#### 4.3.2. Mutations in Complex IV Structural Subunits

In the past two decades, pathogenic mutations that result in COX deficiency have been identified in the structural subunits *COX4*, *COX5A*, *COX6A*, *COX6B*, *COX7B*, *COX8*, and *NDUFA4* (Table 5). With the exception of *COX7B*, patients with described mutations in structural COX subunits are born to consanguineous parents and therefore carry homozygous mutant alleles. Clinical phenotypes include, among others, Leigh or Leigh-like syndrome, encephalopathy, myopathy, and anemia.

Interestingly, *COX7B* is located in the X chromosome and is the only X-linked subunit of COX. Different heterozygous mutations in *COX7B* have been described in patients presenting with microphthalmia with linear skin lesions (MLS), a neurocutaneous X-linked dominant male-lethal disorder [136]. In addition to *COX7B*, MLS has also been associated with mutations in *HCCS*, the holocytochrome *c*-type synthase, and in *NDUFB11*, a subunit of Complex I. Somatic mosaicism and the degree of X chromosome inactivation in different tissues could explain the variability of additional clinical phenotypes that accompany MLS, such as developmental delay, abnormalities of the central nervous system, short stature, cardiac defects, and several ocular anomalies [137]. Indeed, the majority of MLS patients have severe skewing of X chromosome inactivation, probably because during embryonic development, respiratory competent cells multiply faster and outgrow cells harboring mutations in these structural genes [137].

#### 4.3.3. Mutations in Complex IV Assembly Factors

*SURF1*, the human homolog of yeast *SHY1*, has been implicated in the maturation of the heme *a* centers of Complex IV [138]. Mutations in this gene are the most frequent cause of Leigh syndrome stemming from COX deficiency [139]. The first cases of Leigh syndrome caused by *SURF1* mutations were described in 1998 [140,141]. Since then, many other cases have been reported. A systematic review by Wedatilake et al. [139] lists 43 records describing 129 cases of Leigh syndrome with SURF1 deficiency caused by 83 different mutations. The authors also performed a study that included about 50 patients, in which the most frequently occurring mutation was L105* (16 homozygous and 11 compound heterozygous) and no specific correlation of genotype to phenotype was established [139]. Besides Leigh syndrome, there are also reports of *SURF1* mutations associated with Charcot–Marie–Tooth disease [142].

As already mentioned, Cox10 and Cox15 are required for the conversion of heme *b* to the heme *a* of the two redox centers in Cox1 (MTCO1). In *S. cerevisiae*, *cox10* and *cox15* mutants have no Complex IV activity [143,144]. Yeast *cox10* mutants can be functionally complemented by the human homolog of *COX10* [145]. Pathogenic mutations in *COX10* and *COX15* are associated with Leigh syndrome, cardiomyopathy, and encephalopathy, among others (Table 6), and, typically, such patients have an early fatal outcome due to respiratory failure. However, a single adult patient, a 37-year old woman, was identified with isolated COX deficiency associated with a relatively mild clinical phenotype (myopathy, demyelinating neuropathy, premature ovarian failure, short stature, hearing loss, pigmentary maculopathy, and renal tubular dysfunction) due to compound heterozygous mutations resulting in D336V and R339W substitutions in *COX10* [146]. Surprisingly, no COX was detected in blue native gels on mitochondria extracted from the patient’s muscle cells. The mutations were introduced into yeast both individually and in combination, all resulting in the loss of the respiratory capacity, which supported the pathogenicity of the mutations [146].

There is also a single case of a long surviving Leigh syndrome patient resulting from compound heterozygous mutations leading to S152* and S344P substitutions in Cox15. The 16 year old patient presented 42% and 22% of residual COX activity in skeletal muscle cells and fibroblasts, respectively. A normal amount of assembled COX holoenzyme was present in cultured fibroblasts, which could account for the slower clinical progression of the disease [147].

Human SCO1 and SCO2 are copper proteins involved in the metalation of MTCO2 [148]. Although both proteins have homologs in yeast, only Sco1 is needed for metalation of Cox2 [149]. A role of Sco1 in assembly of COX is supported by its presence in a Cox2 assembly intermediate [150]. Pathological mutations in *SCO1* and *SCO2* have been described to result in cardioencephalomyopathy. Gurgel-Giannetti et al. [151] reviewed about 40 patients with mutations in *SCO2*, the majority presenting with cardioencephalomyopathy while two patients suffered from Leigh syndrome. With the exception of one patient that had a homozygous G193S substitution, all other patients presented the E140K substitution which was found either in homozygosis or in association with a second mutation [151]. Furthermore, mutations in *SCO2* have also been associated with Charcot–Marie–Tooth disease [152] and with high degree myopia [153,154].

Additionally, in the past 10 years, pathogenic mutations have been identified in COX assembly factors, products of *FAM36A*, *COX14*, *PET117*, *COA5*, *PET100*, *COA6*, and *COA3* with a variety of clinical phenotypes (Table 5). The pathologies were found in infants with either homozygous or compound heterozygous mutations.

### 4.4. ATP Synthase

The F_1_F_o_-ATP synthase or Complex V is a large multimeric protein complex located in the inner membrane of mitochondria. Its principal function is to phosphorylate ADP to ATP using the energy of the proton gradient formed during oxidation of NADH and succinate [188]. This means of making ATP, historically referred to as oxidative phosphorylation, is reversible, allowing the energy released when the ATP synthase hydrolyzes ATP to be stored as a proton gradient capable of driving other energy-demanding chemical, transport, and physical processes. The mitochondrial ATP synthase consists of three distinct structural components: the F_1_ ATPase, the peripheral stalk, and a membrane-embedded unit referred to as F_o_ [189]. In both mammalian and yeast cells, the F_1_ ATPase is composed of five distinct subunits: three α and three β subunits, that form a hexameric barrel structure, and the monomeric γ, δ, and ε subunits that constitute the central stalk [190]. The peripheral stalk is composed of four nuclear encoded subunits: b, d, OSCP, and h (yeast) or F6 (human). The membrane-embedded domain is comprised of eight subunits, three of which (Atp6, Atp8, and Atp9) are encoded in the mitochondrial genome of *S. cerevisiae* [191], but only two (MT-ATP6 and MT-ATP8) in mammalian mitochondria [192]. Atp9 is present in multiple copies in a ring that rotates during catalysis. The number of Atp9 molecules per ring differs depending on the organism. The yeast and human rings consist of ten and eight subunits of Atp9, respectively [193,194,195].

#### 4.4.1. Mutations in ATP Synthase Mitochondrially Encoded Subunits

The first mitochondrial disease caused by an ATP synthase dysfunction was found in patients with mutations in *MT-ATP6*, a mitochondrial gene encoding a key component of the F_o_ proton channel [196]. The clinical phenotypes of these patients depend largely on the relative levels of heteroplasmy. For instance, patients harboring an m.T8993G substitution with a 70–90% mutation load often exhibit ‘milder’ syndromes, such as NARP (neurogenic muscle weakness, ataxia, and retinitis pigmentosa) or FBSN (familial bilateral striated necrosis) [196]. Patients with the same m.T8993G substitution, but with a mutation load exceeding 90–95%, present with a more severe neurological disorder: maternally inherited Leigh syndrome characterized by fatal infantile encephalopathy [196].

The m.T8993G pathogenic mutation in *MT-ATP6* was the first reported mitochondrial disease associated with an ATP synthase defect [197]. Since then, more than 40 pathogenic variants have been reported [69]. Four of these, m.T8993G, m.T8993C, m.T9185C, and m.T9185C, constitute most of all known cases, while the remaining variants mostly appear as isolated cases [198]. The clinical presentations of these patients are heterogeneous, with Leigh syndrome and NARP being the most frequently reported (Table 7). Recurrent phenotypes also include Charcot–Marie–Tooth peripheral neuropathy, spinocerebellar ataxia, and familiar upper motor neuron disease. Isolated cases of MLASA (mitochondrial lactic acidosis and sideroblastic anemia), HCM, primary lactic acidosis, 3-methylglutaconic aciduria and optic neuropathy have also been reported [198].

Due to the heteroplasmy accompanying mitochondrial deficiencies caused by *MT-ATP6* mutations, the pathogenic mechanisms underlying these disorders have been challenging to elucidate. In this respect, homoplasmic *S. cerevisiae* clones carrying the pathogenic *ATP6* mutations have served as a good model for characterizing the etiology of *MT-ATP6* diseases. The first modeling in yeast of a pathogenic mutation in *ATP6* was done by Rak et al. [199]. These authors have introduced in yeast the equivalent of the T8993G mutation responsible for NARP. Although the ATP synthase was correctly assembled and present at 80% of wild-type levels, the yeast mutant showed poor respiratory growth and mitochondrial ATP synthesis was only 10% of that of the wild-type. The mutant also had lower steady state levels of COX, suggesting a co-regulation of ATP synthase activity and COX expression [199]. Additionally, there is more evidence that the biogenesis of these two enzymes may be coupled. It was recently showed that Atco, an assembly intermediate composed of COX and ATP synthase subunits, namely Cox6 and Atp9, is a precursor and the sole Atp9 source for ATP synthase assembly [200].

Another study showed that a Leigh syndrome causing m.T9191C mutation with an L242P substitution in *ATP6* reduced both assembly of the yeast ATP synthase and the efficiency of ATP synthesis by 90% [201]. Based on the biochemical phenotypes of the mutant and of revertants with amino acid substitutions at this position, the proline was proposed to disrupt the terminal α-helix causing a displacement of other neighboring helices involved in proton transfer at the Atp6 and Atp9 interface [202]. Additionally, the conformational change induced by the mutations promoted proteolysis of Atp6, thereby accounting for the reduction of assembled ATP synthase. Revertants with replacements of the helix-disrupting proline by threonine or serine completely restored the assembly defect, but only partially the efficiency of proton translocation. These observations, combined with modeling of the interface between the two subunits, suggested that the suppressor mutations did not completely compensate for the displacement of residues involved in release of protons from the Atp9 ring and their transfer to a neighboring aspartic acid residue in Atp6 [202]. These findings are consistent with previous models of the disease, suggesting that certain pathogenic *ATP6* mutations do not completely block the proton translocation mechanisms of the F_o_ domain, but rather disrupt proton transport during rotation of the ring. As the conformational changes in the F_1_ moiety are induced by rotation of the Atp9 ring, the uncoupling leads to the inability of ATP synthase to harness the energy normally released from the translocation of protons for ATP synthesis [203].

The different biochemical anomalies discerned in the numerous *MT-ATP6* pathogenic variants suggest several pathophysiological mechanisms responsible for diseases associated with defects in the ATP synthase [198]. These include decreased holoenzyme assembly, destabilization of the proton pore resulting in mitochondrial membrane potential buildup, impairment of the proton pump leading to decreased membrane potential, reduced ATP synthesis and abnormal sensitivity to the ATP synthase inhibitor oligomycin [198].

A smaller subset of mitochondrial mutations causing ATP synthase deficiencies has been ascribed to the second mitochondrial gene, *MT-ATP8,* encoding a subunit of F_o_. The first pathogenic mutation in *MT-ATP8* was identified in a 16-year-old patient with apical HCM and neuropathy, with a marked reduction in ATP synthase activity in fibroblasts and muscle tissue [204]. Since then, several other patients have been identified with pathogenic *MT-ATP8* mutations. One of the patients was a seven-month-old infant diagnosed with tetralogy of Fallot, the most common type of congenital heart defect characterized by ventricular septal defect, pulmonary stenosis, right ventricular hypertrophy and aortic dextroposition [205].

*MT-ATP6* and *MT-ATP8* genes overlap 46 nucleotides that span 16 codons [206]. On the MITOMAP [69] are presently listed four known pathogenic mutations in this region that contribute to an amino acid change in both genes, and three other variants that have a hypothesized deleterious effect. An ATP synthase deficiency has been associated with an m.C8561G mutation in two siblings with cerebral ataxia and loss of neuromuscular function. The mutation, resulting in P12R and P66A changes in *MT-ATP6* and *MT-ATP8*, respectively, when expressed in homoplasmic myoblasts, reduced cellular ATP by 15% but had only a marginal effect on steady state concentrations of the ATP synthase [207]. Even though there was no obvious decrease in the steady-state concentration of ATP synthase, a substantial increase of an uncharacterized assembly intermediate was noted. Another study reported an m.C8561T mutation with other amino acid substitutions in a patient with a similar biochemical but more severe clinical presentation [208]. Neither studies examined the effect of the singular or combined *ATP8* and *ATP6* mutations on the yeast ATP synthase.

#### 4.4.2. Mutations in ATP Synthase Nuclear Structural Subunits

At present, only a small number of mitochondrial pathologies have been attributed to mutations in nuclear genes coding for subunits of the ATP synthase and for chaperones that function in assembly of the enzyme. Several studies found that patients with lactic acidosis, persisting 3-methylglutaconic aciduria, cardiomyopathy, and early death were correlated with severe deficiencies in ATP synthase, but the genetic lesions in these studies were not identified [209,210,211]. *ATP5F1E* and *ATP5F1D*, the human homologs of the yeast *ATP15* and *ATP16*, code for the ε and δ subunits, respectively, of the central stalk of F_1_. A homozygous missense mutation in *ATP5F1E* was the first reported instance of a patient with an ATP synthase defect stemming from a mutation in a nuclear encoded subunit of the enzyme. The mutation caused mild mental retardation and the patient developed peripheral neuropathy [6]. The mitochondrial synthesis of ATP, the oligomycin ATPase activity and the ATP synthase content were reduced by 60–70%. The residual enzyme with the mutated ε subunit had normal activity, indicating that the primary effect of the mutation was on assembly of the synthase. Interestingly, this mutation, when introduced in the ε subunit of yeast ATP synthase, did not elicit any detectable effect on either the activity of assembly of the enzyme [212]. This suggested that the mutation in the human ε subunit, perhaps because of a weaker physical interaction with the c ring, exerted a deleterious effect on assembly of the human but not of the yeast enzyme [212]. In a more recent report, two patients with homozygous mutations in *ATP5F1D* were shown to suffer from metabolic disorders in one case and from encephalopathy in another [213]. *ATP5F1A,* coding for human α subunit of F_1_, is the third nuclear ATP synthase gene identified in two patients presenting fatal neonatal encephalopathy with intractable seizures [5]. Interestingly, reduced levels of cellular ATP5F1 were also shown to correlate significantly with earlier-onset prostate cancer [214]. Indeed, transitioning from oxidative phosphorylation to anaerobic glycolysis for energy production occurs in many types of tumors, and could explain the pathophysiology of the disease.

None of the patients with mutations in the nuclear ATP synthase genes discussed in this and the next sections exhibit the clinical NARP and Leigh syndrome phenotypes characteristic of patients with mutations in the two mitochondrially-encoded genes of the enzyme.

#### 4.4.3. Nuclear ATP Synthase Assembly Gene Mutations

A substantial number of nuclear gene products of *S. cerevisiae* are known to regulate and chaperone different steps of ATP synthase assembly [215]. At present, however, the only known human regulatory factors with identified mutations in a small number of patients are TMEM70, a protein essential for ATP synthase assembly and ATPAF2, the homolog of the yeast Atp12 chaperone that interacts with the α subunit of F_1_ during assembly of this ATP synthase module [216]. TMEM70 does not have a yeast homolog. The phenotypes associated with mutations in TMEM70 and the role of its product in ATP synthase assembly have been reviewed elsewhere [217,218].

A study in which two patients, ascertained to have nuclear mutations that affected mitochondrial ATP synthase assembly, were screened by sequencing human homologs of yeast genes previously shown to affect F_1_ biogenesis led to the identification in one patient of a mutation in *ATPAF2*. This patient, diagnosed with lactic acidosis, glutaconic aciduria, encephalomyopathy and a range of different developmental problems, was found to have a homozygous W94R amino acid substitution that resulted in severe deficits of ATP synthase in the heart, liver and to a lesser degree in skeletal muscle, resulting in death at 14 months of age [219]. The deleterious effect of the W94R substitution in a highly conserved region of *ATPAF2* on ATP synthase assembly was confirmed by Meulemans et al. [7], who showed that the wild-type, but not the mutant human gene, restored the ATP synthase activity of a yeast *atp12* mutant. The *atp12* mutant also failed to be complemented by the yeast *ATP12* harboring the equivalent W102R mutation expressed from a low copy yeast CEN, but not from a high-copy plasmid. The rescue by the W102R, however, depended on the presence of wild-type *FMC1*, a gene implicated in regulating the activity of yeast Atp12 [220].

### 4.5. Coenzyme Q

Coenzyme Q (ubiquinone, CoQ or CoQ_10_ in humans) is a lipophilic redox molecule found in virtually all eukaryotic organisms and most bacteria. It is composed of a quinone ring connected to a polyisoprenoid side chain of variable length. Coenzyme Q serves several crucial functions in mitochondria, including transfer of electrons from Complexes I and II to Complex III, acting as an essential cofactor in the uncoupler protein mediated transfer of protons to the matrix, prevention of lipid peroxidation, biosynthesis of uridine, beta-oxidation of fatty acids, and binding to and regulating the permeability transition of the voltage-dependent anion channel [221].

The CoQ biosynthetic pathway in eukaryotes has been studied in yeast *coq* mutants arrested at different steps of the pathway [222,223]. Like human cells, yeast relies on de novo synthesis of CoQ; and any deficiency in the biosynthetic pathway results in a growth arrest on media containing non-fermentable carbon sources [3,224]. Most enzymes of the CoQ biosynthetic pathway are organized in a multi-subunit complex known as the CoQ synthome [222,225]. The CoQ synthome is spatially linked to the endoplasmic reticulum–mitochondria contact sites, providing optimal CoQ production with an efficient intracellular distribution as well as minimizing the escape of toxic intermediates [226,227]. Expression of functional CoQ in yeast depends on at least 14 nuclear gene products (Coq1-Coq11, Yah1, Arh1, and Hfd1), all located in mitochondria [227].

CoQ biosynthesis is achieved by three separate and highly conserved pathways:

(1) Synthesis of the quinone ring from 4- hydroxybenzoate (4HB), derived from tyrosine [228], or from p-aminobenzoic acid (pABA), in yeast but not in humans [225,227,229]. The early steps of 4HB formation are still to be determined but deamination of tyrosine starts with Aro8 or Aro9 catalyzed transamination [230]. The formation of the final intermediate 4-hydroxybenzaldehyde (4 HBz) is catalyzed by the aldehyde dehydrogenase Hfd1 [230,231].

(2) Synthesis of isopentenyl pyrophosphate (IPP) and dimethylally pyrophosphate (DMAPP), catalyzed by Coq1 in yeast [232] and PDSS1 and PDSS2 in humans.

(3) Prenylation of parahydroxybenzoate by the polisoprenyl transferase Coq2 [233] and further modifications of the benzoquinone ring by hydroxylases and methyl transferases. The isoprenoid side chain is important for proper CoQ localization at the mid-plane of phospholipid bilayers. Yeast coenzyme Q contains 6 isopentenyl units (CoQ_6_) while in humans the major coenzyme Q isoform contains 10 isopentenyl units (CoQ_10_).

The benzoquinone head group is modified by hydroxylations catalyzed by Coq6 and Coq7 [234,235,236], methylation of the resultant hydroxyls by the Coq3 methyl transferase [237], methylation of the ring by Coq5 [238] and a decarboxylation step catalyzed by a still unidentified enzyme of this pathway. Other gene products linked to coenzyme Q biosynthesis and utilization include Coq4 and Coq9, that have been assigned a role in assembly and stability of the CoQ synthome, and Coq8, a member of a protein family that includes kinases and ATP-dependent ligases. Coq8 has been implicated in the phosphorylation state of Coq3, Coq5 and Coq7 [239,240,241]. The steady-state concentrations of Coq4, 6, 7, and 9 are markedly decreased in a *coq8* null mutant, as a result of which assembly of the CoQ synthome is abrogated [242]. The *coq8* null mutant, however, contains a complex of Coq6 and Coq7 thought to be an early intermediate of the synthome [243]. Coq10 is a low molecular weight member of the START protein family that binds coenzyme Q. Although Coq10 is required for respiration, its synthesis is only partially affected in log but not in stationary phase yeast cells, suggesting that its function is related to the delivery of coenzyme Q from the synthome located at the endoplasmic-mitochondrial contact sites to the regions of the inner membrane containing the respiratory chain complexes [226,244]. The respiratory deficiency of *coq10* mutants is partially rescued in a *coq11* mutant, which codes for a component of the synthome that has been proposed to down- regulate the synthesis of coenzyme Q [244]. Coq11 (human NDUFA9), a separate protein in yeast, is a component of the CoQ synthome [245] and appears to regulate its formation and stability [244]. CoQ yeast and human genes are listed in Table 8.

#### Mutations in COQ Genes

CoQ_10_ deficiency, a biochemical lesion first described over three decades ago by Ogasahara et al. [246], is subdivided into primary CoQ_10_ deficiency, when caused by a pathogenic mutation in one of the genes required for the coenzyme’s biosynthesis, and secondary CoQ_10_ deficiency, when the mutated gene is not directly related to the biosynthetic pathway [247]. Of the two, the latter has been reported more frequently in patients, with phenotypes including mitochondrial myopathies, mitochondrial DNA depletion syndrome and multiple acyl-CoA dehydrogenase deficiency (MADD) [248]. However, the pathogenic mechanisms linking these disorders with the observed CoQ_10_ deficiency have yet to be elucidated.

Primary CoQ_10_ deficiency is far rarer [248]. Emmanuele et al. [247] first classified the clinical manifestations of primary CoQ_10_ deficiency into five distinct phenotypes: encephalomyopathy, isolated myopathy, nephropathy, infantile multisystemic disease, and cerebellar ataxia. However, it has been argued that this subdivision should be updated, as new cases have been discovered with novel mutations presenting a wide range of other clinical phenotypes, as well as different combinations of the previously described symptoms [248]. To date, mutations in ten genes have been associated with primary CoQ_10_ deficiency (Table 9).

With the exception of Coq3, patients have been reported with mutations in all other components of the CoQ multi-subunit complex [249,250]. These patients can be treated with CoQ_10_ supplementation with partial success. Early treatment based on early diagnosis is critical for the best outcome [251]. Because of its poor solubility, CoQ_10_ is only administrated in oral formulations, despite its destitute bioavailability [252,253]. Similarly, uptake of CoQ6 in yeast *coq* mutants is inefficient [254,255].

Due to the striking homology between human and yeast COQ genes [227], studies of CoQ proteins in *S. cerevisiae* may provide insight into human homologs, leading to the identification of residues critical for protein function and, therefore, with higher pathogenic potential [256]. Indeed, yeast *coq3*, *coq8*, *coq9*, and *coq10* mutants are complemented by the human counterparts [239,257,258,259], while yeast *coq5* null mutants are complemented by human *COQ5* combined with overexpression of *COQ8* [260]. Furthermore, studies of pathogenic mutations in *COQ* genes have been validated in yeast *coq1* [261], *coq2* [261,262], *coq4* [263], *coq6* [264], *coq8* [241], and *coq9* mutants [265].

*PDSS1* and *PDSS2*, both human homologs of yeast *COQ1*, encode two proteins that form a heterotetramer that catalyzes the elongation of the isoprenoid side chain. *PDSS1* does not complement the yeast *coq1* null mutant [261]. Mutations in both of these genes lead to infantile multisystemic disease, a heterogeneous disorder characterized by psychomotor regression, encephalopathy, optic atrophy, retinopathy, hearing loss, renal dysfunction and heart valvulopathy [247,261]. Mutations in *PDSS2* have been associated with additional phenotypes, including Leigh syndrome, steroid resistant nephrotic syndrome (SRNS)—an atypical manifestation for other mitochondrial disorders but quite common for CoQ_10_ deficiencies; and hepatocellular carcinoma [247,266,267]. It has been shown that the downregulation of *PDSS2* can induce a shift from aerobic metabolism to anaerobic glycolysis, as well as increased chromosomal instability—a possible pathogenic mechanism for hepatocellular carcinoma [267].

Pathogenic mutations in *COQ2*, encoding a parahydroxybenzoate-polyprenyltransferase that catalyzes the addition of the isoprenoid chain to the benzoquinone ring, were the first to be associated with primary CoQ_10_ deficiency [268]. Pathogenic mutations in human *COQ2* have been confirmed in yeast by complementation studies of the yeast *coq2* null mutant [261,262]. Clinical manifestations of these mutations include isolated SRNS, SRNS with encephalomyopathy resembling MELAS, fatal infantile multisystemic disease, and late-onset multiple-system atrophy and retinopathy [247,248].

The first pathogenic mutation in *COQ4*, required for the stability of the CoQ synthome, was found as a haploinsufficiency, with a phenotype similar to that of a heterozygous yeast mutant [263]. The patient presented facial dysmorphism and muscle hypotonia, which improved significantly with CoQ_10_ supplementation [263]. Since then, a total of 19 patients, all infants, have been identified with mutations in *COQ4*, with clinical phenotypes that included cerebellar atrophy, lactic acidosis, seizures, muscle weakness, cardiomyopathy, ataxia, and Leigh syndrome [269,270]. None of the patients suffered from nephropathy, typically found in primary CoQ_10_ deficiency.

To date, only three cases of pathogenic mutations in the methyltransferase encoded by *COQ5*, have been recorded. The affected individuals were three female siblings presenting with non-progressive cerebellar ataxia, dysarthria, and mild to moderate cognitive disability [250]. Two of the three siblings exhibited myoclonic jerks and generalized tonic-clonic seizures in adolescence and early 20s. Next-generation sequencing identified a tandem duplication of the last four exons of *COQ5*, while biochemical studies showed a 33% reduction of CoQ_10_ in skeletal muscle—sufficient to sustain a basal rate but insufficient to reach maximal efficiency of respiration [250].

Mutations in *COQ6*, encoding an enzyme involved in hydroxylation and deamination reactions during CoQ biosynthesis, have been primarily associated with SRNS [264], characterized by significant proteinuria with resulting hypoalbuminemia and edema and presenting with focal segmental glomerulosclerosis [248]. The pathogenicity of the first six *COQ6* mutations in human patients was confirmed by complementation studies using the yeast *coq6* null mutant [264]. Three patients harboring a *COQ6* mutation also suffered from infantile multisystemic disease [247]. Based on in vitro and in vivo studies of renal podocyte cell lines, it has been hypothesized that the pathogenicity of *COQ6* mutations relates to respiratory chain deficiency, ROS generation, disruption of podocyte cytoskeleton and induction of cellular apoptosis, ultimately resulting in SRNS [271].

Three cases of primary CoQ_10_ deficiency caused by mutations in *COQ7*, responsible for the penultimate step of CoQ biosynthesis, have been reported in two children with similar phenotypes of spasticity, sensorineural hearing loss and muscle hypotonia; and a third more severe case of fatal mitochondrial encephalo-myo-nephro-cardiopathy, persistent lactic acidosis, and basal ganglia lesions [272,273,274]. In one of the patients, treatment with the unnatural biosynthesis precursor 2,4-dihydroxybenzoate (DHB), a hydroxylated variant of the native 4-hydroxybenzoic acid (4-HB) normally modified by COQ7, increased CoQ_10_ levels and partially restored mitochondrial respiration [273].

The human genes *ADCK3* and *ADCK4*, also known as *COQ8A* and *COQ8B*, are both homologs of yeast *COQ8*. Mutations in *COQ8A* have mostly been associated with autosomal recessive progressive cerebellar ataxia (ARCA), often accompanied by childhood onset cerebellar atrophy, with and without seizures, and exercise intolerance [241,248,275]. The pathogenic nature of the *ADCK3* mutations was corroborated using the yeast counterpart system [241]. However, isolated cases of psychiatric disorders, seizures, migraines, and dysarthria have been reported [248]. Studies of fibroblast cell lines isolated from ARCA patients showed an increased sensitivity to oxidative stress induced by hydrogen peroxide, high levels of oxidative stress and changes in mitochondrial homeostasis as a result of loss-of-function mutations in *COQ8A* [275]. Interestingly, concomitant with the upregulation of ROS production, increased respiratory supercomplex stability and basal respiratory rate were observed, suggesting that the loss of ADCK3 could result in a compensatory elevation of respirasome formation [275]. The relationship between the two human paralogs *COQ8A* and *COQ8B* is unclear. However, pathogenic mutations in the two genes lead to completely different clinical phenotypes. Indeed, all patients with *COQ8B* mutations suffered from SNRS, with only a single case of neurological involvement reported [248,276].

Primary CoQ_10_ deficiency caused by mutations in *COQ9* is extremely rare, with only seven cases from four families having been reported. The *COQ9* gene product binds to the polyprenyl tail of CoQ intermediates with high specificity, allowing the modification of the benzene ring by Coq7 and other components of the CoQ synthome [259,277]. The reported cases include an infant suffering from lethal lactic acidosis, seizures, cerebral atrophy, HCM, and renal dysfunction [248,265], a boy diagnosed with neonatal Leigh-like syndrome who died at 18 days of age from cardio-respiratory failure [278], four siblings with an unknown and ultimately lethal condition characterized by dilated cardiomyopathy, anemia, abnormal appearing kidney, and suspected Leigh syndrome [279]; and a nine-month old girl presenting with microcephaly, truncal hypotonia, and dysmorphic features [280].

It has been shown that the pathogenicity of primary and secondary CoQ_10_ deficiencies is linked to the impairment of electron transfer to Complex III, ultimately resulting in decreased mitochondrial respiration and ATP synthesis [248,281]. It has also been noted that the neurological presentation of CoQ_10_ deficiency is likely associated with oxidative damage and caspase-independent apoptotic cell death in the brain, as a result of mitochondrial impairment [281]. While renal dysfunction is not uncharacteristic of mitochondrial deficiencies, the glomerular involvement, as opposed to the tubular damage seen in other mitochondrial cytopathies, is perplexing and could be a result of impaired CoQ_10_ antioxidant function [248]. Equally puzzling is the remarkable diversity of clinical phenotypes resulting from mutations in different COQ genes. Based on in vivo studies of mouse models harboring *COQ9* mutations, Luna-Sánchez et al. [281] have hypothesized that a key factor in determining the degree of severity and particular clinical phenotypes of CoQ_10_ deficiencies is the stability of the CoQ synthome. Lastly, decreased CoQ_10_ levels impair the activity of sulfide:quinone oxidoreductase, an enzyme involved in the catabolism of H_2_S [282]. While at physiological conditions H_2_S serves as an electron donor to the mitochondrial respiratory chain, at elevated levels it inhibits Complex IV activity, resulting in reduced cellular respiration. Over-physiological levels of H_2_S could therefore participate in the pathogenic mechanism of CoQ_10_ deficiency [282].

### 4.6. Cytochrome c

Cytochrome *c*, a low-molecular weight heme protein loosely tethered to the inner mitochondrial membrane, is an important component of the respiratory chain that accepts electrons from Complex III and transfers them to Complex IV. Cytochrome *c* also functions as an initiator of apoptosis. In humans, cytochrome *c* is encoded by *CYCS* and in yeast by two isoforms *CYC1* and *CYC7*, the first one being predominant. To date, four mutations in *CYCS* have been associated with non-syndromic and mild thrombocytopenia, a rare autosomal dominant disorder characterized by low platelet levels in the blood but no other hematological findings. These mutations include a G41S substitution found in a New Zealander family [283], a Y48H mutation in an Italian family [284], an A52V mutation in a British family [285], and a K101del deletion in a 64-year-old woman from a Japanese family with multiple cases of Hemophilia A [286]. Unlike other non-syndromic thrombocytopenias, the morphology of circulating platelets in these patients was normal [287]. Some of these mutations, recreated using yeast model systems, led to reduced cytochrome *c* expression, reduced respiratory rate, and increased apoptotic rate [284,286]. However, the link between loss-of-function mutations in *CYSC* and abnormal platelet formation has yet to be elucidated.

The addition of the heme moiety to cytochrome *c* is catalyzed by a cytochrome *c* type heme lyase encoded by the *HCCS* gene, the human homolog of yeast *CYC3*. HCCS is able to complement the yeast Cyc3 deficiency [288]. Mutations in *HCCS* have been associated with MLS, an X-linked, male-lethal disorder described in the Complex III section [289]. It has been hypothesized that the characteristic phenotype of MLS patients is a consequence of the activation of non-canonical, caspase-9 mediated cell death in the brain and eyes as a result of HCCS impairment [290].

## 5. Human Pathologies Resulting from Mutations in Genes with No Homologs in *S. cerevisiae*

### 5.1. Complex I

While *S. cerevisie* mitochondria contain three different single subunit NADH dehydrogenases in the inner membrane [291], the human counterpart forms a large hetero-oligomeric complex (Complex I) comprised of 38 nuclear-encoded and seven mitochondrially encoded subunits. As such, the genes for the numerous structural proteins and assembly factors related to Complex I have no yeast homologs. The majority of Complex I genes have been associated with mitochondrial disorders. Complex I deficiency is the most common biochemical lesion identified in childhood-onset mitochondrial diseases, accounting for approximately 30% of all cases [292]. The clinical presentations of these disorders are varied, including Leigh syndrome, fatal infantile lactic acidosis, HCM, exercise intolerance, LHON, leukoencephalopathy, and MELAS [292]. Although divergent, yeast NADH-dehydrogenase Ndi1 has been tested as a candidate for gene therapy for human Complex I deficiencies, with success in preventing ischemia-reperfusion injury in transgenic rats [293,294].

Other yeasts, such the obligate aerobic *Yarrowia lipolytica*, however, have Complex I and have been used to study and model diseases of this complex [295,296]. In one study, modeling mutations in *Y. lipolityca* made it possible to determine the most pathogenic mutation among two mutated alleles. The compound heterozygous Y53C and Y308C substitutions in *NDUFS2* were found in siblings presenting with non-syndromic LHON-like disease but normal levels of Complex I [297]. The parents, each carrying one of the mutations, presented no clinical phenotype. Equivalent substitutions (H57C and Y311C) were introduced independently in the *Y. lipolytica* homolog *NUCM*, resulting in normal levels of Complex I in mutant H57C and no detectable Complex I in mutant Y311C. The authors concluded that the normal levels of Complex I in the patient’s fibroblast arose from the expression of the Y53C allele and that the non-syndromic LHON-like phenotype was possibly caused by an instability of Complex I containing this mutation [297]. In another study, 13 out of 16 single amino acid substitutions of NDUFV1 were confirmed as pathogenic using *Y. lipolytica* [298]. This yeast is also useful in the evaluation of pathogenic mutations in Complex I assembly factors, such as NUBPL (nucleotide binding protein-like), which were recreated in the homologous Ind1 protein of the yeast model [299]. Recently, the cryo-EM structure of *Y. lipolytica* Complex I has been solved [300,301] and can provide helpful information for those modeling Complex I diseases in this yeast.

### 5.2. Assembly Factors

Pathogenic mutations have also been described for some human genes with no yeast homologs encoding assembly factors of oxphos complexes, such as *TCC19* (Complex III), *TACO1* (Complex IV), and *TMEM70* (ATP synthase). They are beyond the scope of this review.

## 6. Concluding Remarks

Mammalian mitochondrial DNA codes for 13 proteins, all of which are subunits of the respiratory complexes and ATP synthase. Most of the organelle, however, is encoded in the nuclear genome, by what, in yeast, is estimated to be at least 900 genes or 15% of the genome. It stands to reason that a large number of human genetic disorders will stem from mutations that affect mitochondrial function. In the past, studies of yeast *pet* mutants have had a direct impact on understanding the genetic and mechanistic basis of human mitochondrial diseases. However, about 20% of the eukaryotic proteome has not yet been characterized, even in well-studied model organisms such as *S. cerevisiae* [302]. A substantial number of proteins still lacking an ascribed function are encoded by genes that affect mitochondrial respiration. In this context, yeast is still a powerful platform for discovering the function of uncharacterized mitochondrial proteins, as well as to gain a better understanding of the underlying molecular consequences of pathogenic mutations that could prove to be promising targets for mitochondrial disease therapies.

## Figures and Tables

**Figure 1 life-10-00304-f001:**
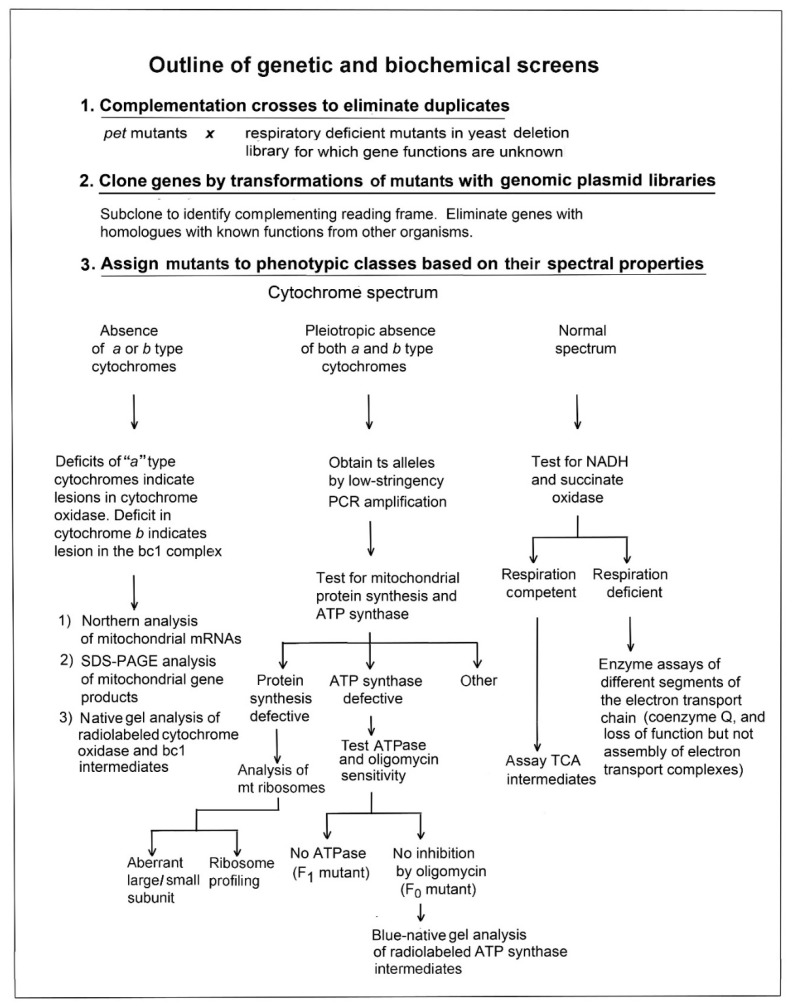
Genetic and biochemical screening of *pet* mutants. The initial complementation tests are done to identify *pet* mutants with counterparts in the knockout strain collection. This eliminates the need to clone and sequence genes already annotated.

**Table 1 life-10-00304-t001:** Pathologies resulting from mutations in genes encoding Complex II subunits and assembly factors, and their yeast homologs.

Human Gene	Yeast Gene	Clinical Phenotype ^1^	Mutation	Confirmation in Yeast	Reference
*SDHA*	*SDH1*	Leigh syndrome	homozygous R554W	yes	[25]
			compound heterozygous A524V, M1L	no	[45]
			compound heterozygous W119*, A83V	no	[46]
			homozygous G555E	no	[27]
		late-onset optic atrophy, ataxia, myopathy	heterozygous R408C	no	[47]
		neonatal isolated cardiomyopathy	homozygous G555E	no	[28]
		undefined ^2^	homozygous G555E	no	[29]
		encephalopathy	compound heterozygous—stop codons at residues 56 and 81	no	[48]
		cardiomyopathy, leukodystrophy	compound heterozygous T508I, S509L	no	[49]
		optic atrophy, progressive, polyneuropathy, cardiomyopathy	heterozygous R451C	no	[50]
*SDHB*	*SDH2*	hypotonia, leukodystrophy	homozygous D48V	yes	[49]
		leukoencephalopathy	compound heterozygous D48V, R230H; homozygous L257V ^3^	no	[51]
*SDHD*	*SDH4*	early progressive encephalomyopathy	compound heterozygous E69K, *164Lext*3	no	[52]
*SDHAF1*	*SDH6*	leukoencephalopathy	homozygous G57R; homozygous R55P ^3^	yes	[34]
			homozygous R55P; homozygous Q8*; homozygous G57E ^3^	no	[37]

^1^ Paragangliomas were not included in the table. ^2^ The patient died at five months of age following a respiratory infection before developing other phenotypes. ^3^ In different patients.

**Table 2 life-10-00304-t002:** Yeast complex III subunits and their human homologs. The table also shows Complex III assembly factors that are associated with diseases.

	Yeast Gene	Genome	*pet* Mutant	Human Gene	Function
Enzyme Subunits	*COB*	mitochondrial	N/A	*MT-CYB*	catalysis
	*CYT1*	nuclear	yes	*CYC1*	catalysis
	*RIP1*	nuclear	yes	*UQCRFS1*	catalysis
	*COR1*	nuclear	yes	*UQCRC1*	structure
	*COR2*	nuclear	yes	*UQCRC2*	structure
	*QCR6*	nuclear	no	*UQCRH*	structure
	*QCR8*	nuclear	yes	*UQCRQ*	structure
	*QCR7*	nuclear	yes	*UQCRB*	structure
	*QCR9*	nuclear	no	*UQCR10*	structure
	*QCR10*	nuclear	no	*UQCR11*	structure
Assembly Factors	*CBP3*	nuclear	yes	*UQCC1*	translation/assembly of cyt. b
	*CBP6*	nuclear	yes	*UQCC2*	translation/assembly of cyt. b
	*CBP4*	nuclear	yes	*UQCC3*	assembly of cyt. b
	*MZM1*	nuclear	no	*LYRM7*	assembly of Rieske protein
	*BCS1*	nuclear	yes	*BCS1L*	assembly of Rieske protein

**Table 3 life-10-00304-t003:** Pathologies resulting from mutations in genes encoding Complex III subunits and assembly factors, and their yeast homologs.

Human Gene	Yeast Gene	Clinical Phenotype	Mutation	Confirmation in Yeast	Reference
*MT-CYB*	*COB*	MELAS, LHON, hearing loss, exercise intolerance	many		[70] ^1^
*UQCR4*	*CYT1*	ketoacidosis and insulin-responsive hyperglycemia	homozygous L215F; homozygous T96C ^2^	yes	[115]
*UQCRFS1*	*RIP1*	cardiomyopathy and alopecia totalis	homozygous V72_T81del10; heterozygous V14D; heterozygous R204∗ ^2^	no	[116]
*UQCRC2*	*COR2*	neonatal onset of hypoglycemia	homozygous R183W	no	[77,78]
*UQCRB*	*QCR7*	hypoglycemia	homozygous 4 bp deletion at nucleotides 338–341	yes [75]	[74]
*UQCRQ*	*QCR8*	severe psychomotor retardation, dystonia, athetosis, ataxia, dementia	homozygous S45F	no	[76]
*UQCC2*	*CBP6*	intrauterine growth retardation, renal tubular dysfunction	homozygous c.214-3C>G ^3^	no	[111]
		neonatal encephalomyopathy	homozygous R8P and L10F ^4^	no	[112]
*UQCC3*	*CBP4*	hypoglycemia, hypotonia, delayed development	homozygous V20E	no	[114]
*BCS1L*	*BCS1*	GRACILE syndrome, Björstand syndrome, encephalopathy, muscle weakness	many		see main text
*LYRM7*	*MZM1*	early onset severe encephalopathy	homozygous D25N	yes	[106]
		leukoencephalopathy	homozygous c.243_244 + 2del ^3^	no	[107]
			several	yes	[108]
			homozygous 4bp deletion c.[243_244 + 2delGAGT] ^3^	no	[109]
		liver dysfunction	homozygous R18Dfs*12	no	[110]

^1^ Review describing many mutations. ^2^ In different patients. ^3^ Mutation causing a splicing defect. ^4^ In the same patient.

**Table 4 life-10-00304-t004:** Yeast complex IV subunits and their human homologs. The table also shows Complex IV assembly factors that are associated with diseases.

	Yeast Gene	Genome	*pet* Mutant	Human Gene	Function
Enzyme Subunits	*COX1*	mitochondrial	N/A	*MTCO1*	catalysis
	*COX2*	mitochondrial	N/A	*MTCO2*	catalysis
	*COX3*	mitochondrial	N/A	*MTCO3*	structure/catalysis
	*COX5a*	nuclear	yes	*COX4*	structure
	*COX5b* ^1^	nuclear	no	*-*	structure
	*COX6*	nuclear	yes	*COX5A*	structure
	*COX4*	nuclear	yes	*COX5B*	structure
	*COX13*	nuclear	no	*COX6A*	structure
	*COX12*	nuclear	yes	*COX6B*	structure
	*COX9*	nuclear	yes	*COX6C*	structure
	*COX7*	nuclear	yes	*COX7A*	structure
	*-*	nuclear	-	*COX7B*	structure
	*COX8*	nuclear	no	*COX7C*	structure
	*-*	nuclear	-	*COX8*	structure
	*-*	nuclear	-	*NDUFA4/COXFA4*	structure
Assembly Factors	*COX20*	nuclear	yes	*COX20/FAM36A*	membrane insertion of Cox2
	*COX14*	nuclear	yes	*COX14*	regulation of *COX1* expression, maintenance of monomeric Cox1
	*PET117*	nuclear	yes	*PET117*	couples synthesis of heme *a* to COX assembly
	*PET191*	nuclear	yes	*COA5*	required for COX assembly
	*PET100*	nuclear	yes	*PET100*	required for COX assembly
	*COA6*	nuclear	no	*COA6*	maturation of the Cu_A_ site
	*COX25/COA3*	nuclear	yes	*COA3*	translational regulation of *COX1* mRNA
	*COX15*	nuclear	yes	*COX15*	conversion of heme *o* to heme *a*
	*COX10*	nuclear	yes	*COX10*	farnesylation of heme *b*
	*SCO1*	nuclear	yes	*SCO1*	maturation of the Cu_A_ site
	*SCO2*	nuclear	no	*SCO2*	maturation of the Cu_A_ site
	*SHY1*	nuclear	yes	*SURF1*	hemylation of Cox1

^1^ Yeast subunit 5b is a paralog of subunit 5a and under standard conditions of growth is present at low concentrations [134].

**Table 5 life-10-00304-t005:** Pathologies resulting from mutations in genes encoding Complex IV subunits and assembly factors, and their yeast homologs.

Human Gene	Yeast Gene	Clinical Phenotype	Mutation	Confirmation in Yeast	Reference
*MTCO1*	*COX1*	prostate cancer, LHON, SNHL, DEAF	many		See main text
*MTCO2*	*COX2*	progressive encephalomyopathy, HCM, SNHL, DEAF, LHON	many		See main text
*MTCO3*	*COX3*	LHON, Alzheimer’s disease	many		See main text
*COX4*	*COX5a*	pancreatic insufficiency, dyserythropoeitic anemia, calvarial hyperostosis	homozygous E138K	no	[155]
		short stature, poor weight gain, mild dysmorphic features with highly suspected Fanconi anemia	homozygous K101N	no	[156]
*COX5A*	*COX6*	early-onset pulmonary arterial hypertension, failure to thrive	homozygous R107C	no	[157]
*COX6a*	*COX13*	axonal or mixed form of Charcot-Marie-Tooth disease	homozygous c.247−10_247−6delCACTC ^3^	no	[158,159]
*COX6b*	*COX12*	infantile encephalomyopathy, myopathy, growth retardation	homozygous R19H	yes	[160]
		encephalomyopathy, hydrocephalus, HCM	homozygous R20C	no	[161]
		cystic leukodystrophy	homozygous c.241A>C	no	[162]
*COX7b*		microphthalmia with linear skin lesions	heterozygous c.196delC; heterozygous c.41-2A>G ^3^; heterozygous Q19∗ ^2^	no	[136]
*COX8*		Leigh-like syndrome, epilepsy	homozygous c.115-1G>C ^3^	no	[163]
*NDUFA4*		Leigh syndrome	homozygous c.42+1G>C ^3^	no	[164]
*FAM36A*	*COX20*	dystonia and ataxia	homozygous T52P	no	[165,166]
		dysarthria, ataxia, sensory neuropathy	compound heterozygous K14R, G114S, c.157+3G>C ^3^	no	[167]
		axonal neuropathy, static encephalopathy	compound heterozygous L14R, W74C	no	[168]
*COX14*	*COX14*	fatal neonatal lactic acidosis, dysmorphic features	homozygous M19I	no	[169]
*PET117*	*PET117*	neurodevelopmental regression, medulla oblongata lesions	homozygous c.172C>T (stop codon at residue 58)	no	[170]
*COA5*	*PET191*	fatal neonatal cardiomyopathy	homozygous A53P	no	[171]
*PET100*	*PET100*	Leigh syndrome	homozygous c.3G>C (p.Met1?)	no	[172]
		fatal infantile lactic acidosis	homozygous Q48*	no	[173]
*COA6*	*COA6*	HCM	homozygous W66R	no	[174]
			compound heterozygous W59C, E87*	yes [175]	[162]
*COA3*	*COX25*	neuropathy, exercise intolerance, obesity, short stature	compound heterozygous L67Pfs*21, Y72C	no	[176]
*COX15*	*COX15*	Leigh syndrome	homozygous L139V	no	[177]
			homozygous R217W	no	[178]
			compound heterozygous S152*, S344P	no	[147]
		infantile cardioencephalopathy	compound heterozygous S151*, R217W	no	[179]
		early-onset fatal HCM	compound heterozygous R217W, c.C447-3G ^3^	no	[180]
*COX10*	*COX10*	severe muscle weakness, hypotonia, ataxia, ptosis, pyramidal syndrome, status epilepticus	homozygous N204K	yes	[8]
		Leigh-like disease	homozygous T>C in the ATG start codon	no	[181]
		anemia, sensorineural deafness, fatal infantile HCM	compound heterozygous T196K, P225L	no	[182]
		Leigh syndrome, anemia	compound heterozygous D336V, D336G	no	[182]
		myopathy, demyelinating neuropathy, premature ovarian failure, short stature, hearing loss	compound heterozygous D336V, R339W	yes	[146]
		Leigh syndrome, anemia	homozygous P225L	no	[183]
		hypotony, sideroblastic anemia, progressive encephalopathy	compound heterozygous M344V, L424Pfs	no	[183]
		developmental delay, short stature	compound heterozygous R228H, deletion disrupting the last 2 exons	no	[184]
*SCO1*	*SCO1*	encephalopathy, hepatopathy, hypotonia, or cardiac involvement	homozygous G106del	no	[185]
		fatal infantile encephalopathy	compound heterozygous M294V, V93*	no	[186]
		early onset HCM, encephalopathy, hypotonia, hepatopathy	homozygous G132S	no	[187]
		neonatal-onset hepatic failure, encephalopathy	compound heterozygous P174L, ΔGA nt 363–364	no	[8]
*SCO2*	*SCO2*	cardioencephalomyopathy, Leigh syndrome, high myopia, Charcot-Marie-Tooth disease ^2^	many		[151,152,154] ^1^
*SURF1*	*SHY1*	Leigh syndrome, Charcot-Marie-Tooth disease ^2^	many		[139,142] ^1;^

^1^ Review describing many mutations. ^2^ In different patients. ^3^ Splicing mutation.

**Table 6 life-10-00304-t006:** Human ATP Synthase subunits and theirs yeast homologs.

ATP Synthase Sector	Human Subunit	Human Gene	Genome in Human	Yeast Subunit	Yeast Gene	Genome in Yeast	*pet* Mutant
F_1_ catalytic barrel	α	*ATP5F1A*	nuclear	α	*ATP1*	nuclear	yes
	β	*ATP5F1B*	nuclear	β	*ATP2*	nuclear	yes
F_1_ central stalk	γ	*ATP5F1C*	nuclear	γ	*ATP3*	nuclear	yes
	ε	*ATP5F1E*	nuclear	ε	*ATP15*	nuclear	yes
	δ	*ATP5F1D*	nuclear	δ	*ATP16*	nuclear	yes
Peripheral stalk	b	*ATP5PB*	nuclear	b	*ATP4*	nuclear	yes
	d	*ATP5PD*	nuclear	d	*ATP7*	nuclear	yes
	OSCP	*ATP5PO*	nuclear	OSCP	*ATP5*	nuclear	yes
	F_6_	*ATP5PF*	nuclear	h	*ATP14*	nuclear	yes
F_o_ rotor	Atp6	*MT-ATP6*	mitochondrial	Atp6	*ATP6*	mitochondrial	N/A
	Atp8	*MT-ATP8*	mitochondrial	Atp8	*ATP8*	mitochondrial	N/A
	c1	*ATP5MC1*	nuclear	Atp9	*ATP9*	mitochondrial	N/A
	c2	*ATP5MC2*	nuclear				
	c3	*ATP5MC3*	nuclear				
F_o_ supernumerary	f	*ATP5MF*	nuclear	f	*ATP17*	nuclear	yes
	6.8 PL	*ATP5MPL*	nuclear	i/j	*ATP18*	nuclear	no
	DAPIT	*ATP5MD*	nuclear	k	*ATP19*	nuclear	no
	g	*ATP5MG*	nuclear	g	*ATP20*	nuclear	no
	e	*ATP5ME*	nuclear	e	*ATP21*	nuclear	no

**Table 7 life-10-00304-t007:** Pathologies resulting from mutations in genes encoding ATP synthase subunits and assembly factors, and their yeast homologs.

Human Gene	Yeast Gene	Clinical Phenotype	Mutation	Confirmation in Yeast	Reference
*MT-ATP6*	*ATP6*	NARP, FBSN, Leigh syndrome, Charcot-Marie-Tooth disorder, HCM, MLASA	many		[198] ^1^
*MT-ATP8*	*ATP8*	apical HCM, neuropathy	homoplasmic W55*	no	[204]
		tetralogy of Fallot	homoplasmic G9804A, C8481T, heteroplamic T7501C ^3^	no	[205]
*ATP5F1D*	*ATP16*	metabolic disorders, encephalopathy	homozygous P82L; homozygous V106G ^2^	no	[213]
*ATP5F1E*	*ATP15*	mild mental retardation, developed peripheral neuropathy	homozygous Y12C	no	[6]
*ATP5F1A*	*ATP1*	fatal neonatal encephalopathy	heterozygous R329C	no	[5]
*ATPAF2*	*ATP12*	microcephaly etc	homozygous W94R	yes	[7,211]

^1^ Review describing many mutations. ^2^ In different patients. ^3^ In the same patient.

**Table 8 life-10-00304-t008:** Coenzyme Q genes required for functional expression of CoQ and their yeast homologs.

Yeast Gene	Human Gene
*COQ1*	*PDSS1*
*COQ1*	*PDSS2*
*COQ2*	*COQ2*
*COQ3*	*COQ3*
*COQ4*	*COQ4*
*COQ5*	*COQ5*
*COQ6*	*COQ6*
*COQ7/CAT5*	*COQ7*
*COQ8/ABC1*	*COQ8A*
*COQ8/ABC2*	*COQ8B*
*COQ9*	*COQ9*
*COQ10*	*COQ10A*, *COQ10B*
*COQ11*	*NDUFA9*

**Table 9 life-10-00304-t009:** Pathologies resulting from mutations in Coenzyme Q genes and their yeast homologs.

Human Gene	Yeast Gene	Clinical Phenotype	Mutation	Confirmation in Yeast	Reference
*PDSS1*	*COQ1*	infantile multisystemic disease	homozygous D308E	yes	[261]
*PDSS2*	*COQ1*	infantile multisystemic disease, SRNS, LS, hepatocellular carcinoma	compound heterozygous Q322*, S382L	yes	[262]
*COQ2*	*COQ2*	SRNS, SRNS with encephalomyopathy resembling MELAS, fatal infantile multisystemic disease	homozygous Y297C	yes	[268]
*COQ4*	*COQ4*	LS, cerebellar atrophy, lactic acidosis, etc.	many		[263,269,270] ^1^
*COQ5*	*COQ5*	cerebellar ataxia, dysarthria, myoclonic jerks	biallelic 9590 bp duplication	no	[250]
*COQ6*	*COQ6*	SRNS, infantile multisystemic disease	many		[264] ^1^
*COQ7*	*COQ7*	spasticity, sensorineural hearing loss etc.	homozygous V141E; homozygous L111P; compound heterozygous R107W, K200Ifs*56 ^2^	no	[272,273,274]
*COQ8A*	*COQ8*	ARCA, seizures, dystonia, spasticity	many		[248] ^1^
*COQ8B*	*COQ8*	SRNS	many		[248,276] ^1^
*COQ9*	*COQ9*	infantile multisystemic disease, Leigh-like Syndrome, microcephaly	many		See main text

^1^ Review describing many mutations. ^2^ In different patients.

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
