# Peer review of "Human Mitochondrial Pathologies of the Respiratory Chain and ATP Synthase: Contributions from Studies of Saccharomyces cerevisiae"

_life, 2020, doi:10.3390/life10110304_

Round 1
Reviewer 1 Report
This review by Franco and colleagues clearly shows the usefulness of using yeast as a model organism to study human mitochondrial diseases. There are succinct descriptions of the yeast respiratory chain complexes II, III, IV, the ATP synthase, and the synthesis of CoQ. Figure 1 summarizes the genetic and biochemical methods for screening pet mutants that have contributed in understanding the function and assembly of these complexes and to better understand disease mechanisms. The authors nicely combine clinical phenotypes of mitochondrial diseases in humans with specific mutations in the different components of the respiratory chain and ATP synthase with their counterparts in yeast in the different tables throughout the review. This compilation of data and references will be a useful addition to the literature within the field.
I have a few specific comments regarding the manuscript:
Line 78 – The “assembly of oxphos” complexes is stated but references 5-7 are specifically for ATP synthase subunits, are these the correct references for this sentence?
It is stated that paragangliomas were not included by the authors in this review as mutations leading to this have been reviewed recently; including this as part of the footnote of Table 1 would be helpful when looking at the table.
In Table 2 the assembly factor CYC2 included in the list of assembly factors although it is not discussed in the main text of the review, should this assembly factor be included in the table?
The fifth table in the paper is labeled as Table 6 and referenced as Table 6 within the main text; this table should be labeled as Table 5. This table also has a 4th footnote “In the same patient” which is not used/shown in the table.
Author Response
Reviewer #1
Line 78 – The “assembly of oxphos” complexes is stated but references 5-7 are specifically for ATP synthase subunits, are these the correct references for this sentence?
It has been corrected.
It is stated that paragangliomas were not included by the authors in this review as mutations leading to this have been reviewed recently; including this as part of the footnote of Table 1 would be helpful when looking at the table.
We have added “Paragangliomas were not included in the table” as a footnote in Table 1.
In Table 2 the assembly factor CYC2 included in the list of assembly factors although it is not discussed in the main text of the review, should this assembly factor be included in the table?
It has been removed from Table 2.
The fifth table in the paper is labeled as Table 6 and referenced as Table 6 within the main text; this table should be labeled as Table 5. This table also has a 4th footnote “In the same patient” which is not used/shown in the table.
The mistake in Table 5 label has been corrected. The footnote has been removed.
Reviewer 2 Report
In their manuscript "Human mitochondrial pathologies of the respiratory chain and ATP synthase: contributions from yeast studies", Franco et al. review all the currently known about the genetics and clinical phenotypes of mitochondrial diseases of the respiratory chain and ATP synthase and the yeast modeling. This review is well organized, comprehensive and almost complete (see major points) for the topic. However, in general, the information presented is rarely discussed, which makes this review losing interest.
For example, chapter 3 "Strategy for determining the function of unknown mitochondrial proteins, line 82" describes the process for identifying OXPHOS mutants but no discussion of the approaches used is provided, making this section less interesting. Then follows a listing of the different mutations modeled or not in yeast from complex II to complex V and CoQ. Finally, the last part (chapter 5,
“Human pathologies resulting from mutations in genes with no yeast homologs”, line 803) presents very briefly mutations in genes with no yeast homologs (complex I) while there are several references on yeast models of complex I (e.g. Characterization of clinically identified mutations in NDUFV1, the flavin-binding subunit of respiratory complex I, using a yeast model system, Varghese et al. 2015 Human Molecular Genetics). The authors were limited to present only Saccharomyces cerevisiae, however there are other yeasts (e.g. Yarrowia lipolytica).
However, this review is a good glossary of human OXPHOS mutations modeled in yeast, but lacks a real discussion. Finally, this review also presents some overlap with other reviews in the field.
Major points:
1 - It would be interesting for the authors to present the modeling of pathological mutations of complex I in yeasts other than S. cerevisiae.
2 - In general, it would be interesting for the authors to discuss more in depth the information they present, for example on human mutations that, when modeled in yeast, do not produce the same phenotypes.
Minor points:
1 - The authors give as a reference for modeling in yeast the ATP5F1E mutant (Mayr et al 2010, line 1448), whereas it is Sardin et al 2015, PMID 25954304.
2 - The authors did not present and cite the first modeling in yeast of the ATP6 gene mutation but cite a review (line 495), instead of Rak et al. 2007 PMID 17855363.
3 - In general it will be necessary to check bibliographical references.
Author Response
Reviewer #2
Major points:
1 - It would be interesting for the authors to present the modeling of pathological mutations of complex I in yeasts other than S. cerevisiae.
To indicate to the reader that this review discusses specifically baker’s yeast, the title of the article has been updated to “Human mitochondrial pathologies of the respiratory chain and ATP synthase: contributions from studies of Saccharomyces cerevisiae”. However, we have also mentioned the modeling of pathogenic mutations of Complex I in Y. lipolytica as suggested.
The paragraph bellow has been added to the text (LINE 839):
Other yeasts, such the obligate aerobic Yarrowia lipolytica, however, have Complex I and have been used to study and model diseases in this complex (Kerscher et al, 2002; Kerscher et al, 2004). In one study, modeling mutations in Y. lipolityca made it possible to determine the most pathogenic mutation among two mutated alleles. The compound heterozygous Tyr53Cys and Tyr308Cys substitutions in NDUFS2 were found in siblings presented with non-syndromic LHON-like disease but normal levels of Complex I (Gerber et al, 2017). The parents, each carrying one of the mutations, presented no clinical phenotype. Equivalent substitutions (His57Cys and Tyr311Cys) were introduced independently in the Y. lipolytica homolog NUCM, resulting in normal levels of Complex I in mutant His57Cys and no detectable Complex I in mutant Tyr311Cys. The authors concluded that the normal levels of Complex I in the patient’s fibroblast arose from the expression of the Tyr53Cys allele and that the non-syndromic LHON-like phenotype was possibly caused by an instability of Complex I containing this mutation (Gerber et al, 2017). In another study, 13 out of 16 single amino acid substitutions of NDUFV1 were confirmed as pathogenic using Y. lipolytica (Varghese et al., 2015). This yeast is also useful in the evaluation of pathogenic mutations in Complex I assembly factors, such as NUBPL (nucleotide binding protein-like), which were recreated in the homologous Ind1 protein of the yeast model (Maclean et al., 2018) Recently, the cryo-EM structure of Yarrowia lipolytica Complex I has been solved (Parey et al, 2019; Parey et al, 2019) and can provide helpful information for those modeling Complex I diseases in this yeast.
2 - In general, it would be interesting for the authors to discuss more in depth the information they present, for example on human mutations that, when modeled in yeast, do not produce the same phenotypes.
The paragraph bellow has been added to the text (LINE 82):
Human cells are more complex than yeast cells, the same can be stated about the mitochondria of these two organisms. Higher the organization and the complexity, more different are the consequences of one given deficiency. In many tested mutations, the phenotypes observed in humans are more deleterious for the cell survival than in the yeast counterpart, which is not only true because yeast can ferment but also because of the variable energy demand of a complex organism with different tissues. As an example, the deletion of MRX10 in yeast did not impair its respiratory capacity but mutations in the human counterpart led to respiratory impairment even in cells with low energetic demand such as fibroblasts (Garcia-Diaz et al., 2012). In other circumstances, due to the need of proper protein-protein interactions, or just because of evolutionary divergence, the possibility of heterologous complementation is lost. For instance, yeast shy1 mutants are not complemented by the human homolog SURF1 even if with chimeric versions of the gene (Barrientos, 2003). However, when the human genes do not complement the yeast correspondent mutant, it is still possible to evaluate the pathogenicity of a given mutation by constructing an allele with the corresponding change in the yeast gene.
Minor points:
1 - The authors give as a reference for modeling in yeast the ATP5F1E mutant (Mayr et al 2010, line 1448), whereas it is Sardin et al 2015, PMID 25954304.
It has been corrected.
2 - The authors did not present and cite the first modeling in yeast of the ATP6 gene mutation but cite a review (line 495), instead of Rak et al. 2007 PMID 17855363.
It has been corrected. Rak et al 2007 has been added as the first yeast modeling of a pathogenic ATP6 mutation.
The paragraph bellow has been added to the text (LINE 512):
The first modeling in yeast of a pathogenic mutation in ATP6 was done by Rak et al (2007). These authors have introduced in yeast the equivalent of the T8993G mutation responsible for NARP. Although the ATP synthase was correctly assembled and contained 80% levels of wild type, the yeast mutant showed poor respiratory growth and mitochondrial ATP synthesis was only 10% of that of wild type. The mutant also had lower steady state levels of COX, suggesting a co-regulation of ATP synthase activity and COX expression (Rak et al, 2007). Also, there is more evidence that the biogenesis of these two enzymes may be coupled. It was recently showed that Atco, an assembly intermediate composed of COX and an ATP synthase subunits, namely Cox6 and Atp9, is a precursor and the sole Atp9 source for ATP synthase assembly (Franco et al, 2020).
3 - In general it will be necessary to check bibliographical references.
References have been checked.